# Sublethal and Transgenerational Effects of Cyclaniliprole on Demographic Parameters in *Rhopalosiphum padi* and *Schizaphis graminum* (Hemiptera: Aphididae)

**DOI:** 10.3390/insects16090882

**Published:** 2025-08-25

**Authors:** Xinan Li, Xiaoya Zhang, Wen Zhang, Chengze Song, Fengfan Wang, Ruiyang Qin, Ganyu Zhu, Guochang Wang, Jiangao Yu, Hongliang Wang

**Affiliations:** Henan Province Engineering Research Center of Biological Pesticide & Fertilizer Development and Synergistic Application, School of Plant Protection and Environment, Henan Institute of Science and Technology, Xinxiang 453003, China; lixinan2019@126.com (X.L.); 18455546581@163.com (X.Z.); zihuanrencai@126.com (W.Z.); 15939588659@163.com (C.S.); wangfengfanwff@163.com (F.W.); 15290364144@163.com (R.Q.); 15515255356@163.com (G.Z.); wgchslbh@163.com (G.W.)

**Keywords:** cyclaniliprole, wheat aphids, sublethal concentration, longevity, fecundity

## Abstract

Pest control primarily depends on the application of chemical insecticides. Cyclaniliprole, a novel anthranilic diamide insecticide commercially introduced in 2017, exhibits broad-spectrum activity against various pests, including Hemiptera (aphids, whiteflies), Thysanoptera (thrips), and Lepidoptera (diamondback moth), with particularly remarkable efficacy against resistant populations. *Rhopalosiphum padi* (Linnaeus) and *Schizaphis graminum* (Rondani) (Hemiptera: Aphididae) are the two primary aphid species that infest wheat crops, and these pests frequently occur simultaneously. Nevertheless, the toxicity of cyclaniliprole to *R. padi* and *S. graminum*, along with its sublethal effects on their demographic parameters, remains unclear. Our results indicate that cyclaniliprole shows significantly acute toxicity against *R. padi* and *S. graminum*. Sublethal concentrations significantly reduced adult longevity and reproductive capacity in the F_0_ generation of these wheat aphid species, with no observed hormetic effects in the F_1_ generations of either *R. padi* or *S. graminum*. These results highlight the practical application value of cyromacridine, and support the optimization of integrated wheat aphid management strategies.

## 1. Introduction

Cyclaniliprole is a novel anthranilic diamide insecticide that was commercially launched in 2017 and has since been registered and marketed in multiple countries, including the United States, Canada, Australia, and Brazil [1,2]. As a diamide compound and third-generation ryanodine receptor (RyR) activator, it specifically binds to RyR receptors in insect muscle cells, causing abnormal calcium ion release that leads to muscle paralysis and death [3]. This insecticide demonstrates broad-spectrum efficacy against various pests, including Hemiptera (aphids, whiteflies), Thysanoptera (thrips), and Lepidoptera (diamondback moth), with particularly outstanding performance against resistant populations [2,4]. Its systemic properties provide long-lasting protection while maintaining safety to non-target organisms, earning it classification as a reduced-risk insecticide by the U.S. EPA [5]. Currently, it is primarily used for pest control in high-value crops such as fruits, vegetables, and tea, serving as an important tool in resistance management strategies.

Wheat, a critically important global food crop, faces severe threats from wheat aphids, which rank among the most destructive agricultural pests. These insects contribute to substantial yield losses, accounting for 10–30% of annual global wheat production [6], thereby posing a serious risk to food security worldwide. *Rhopalosiphum padi* (Linnaeus) and *Schizaphis graminum* (Rondani) (Hemiptera: Aphididae) are the two primary aphid species that infest wheat crops, and these pests frequently occur simultaneously [7,8]. These insects inflict direct damage by persistently feeding on phloem sap, depleting essential nutrients and weakening host plants [9]. Beyond sap consumption, they act as vectors for plant viruses, excrete honeydew that promotes sooty mold growth, and impair photosynthetic efficiency, collectively contributing to significant reductions in wheat yield and grain quality [10,11].

The management of wheat aphids primarily relies on chemical insecticides, which are typically applied at optimal concentrations to ensure effective pest control. However, due to rapid pesticide degradation under field conditions and continuous plant growth, residual insecticide concentrations often decline to sublethal levels shortly after application [12]. A growing body of research indicates that sublethal insecticide exposure can induce significant behavioral and physiological modifications in pests, potentially triggering cascading effects on population dynamics, species interactions, and crop damage patterns [13,14,15]. While numerous studies have demonstrated that sublethal concentrations can negatively impact pest physiology by reducing survival rates, impairing growth and fecundity, and disrupting biochemical pathways [7,16], paradoxically, certain sublethal doses may inadvertently enhance pest infestations by stimulating growth and reproduction [17,18,19]. These contrasting findings underscore the critical importance of investigating both the sublethal and transgenerational effects of insecticides for developing sustainable integrated pest management strategies.

In this study, we employ the age-stage two-sex life table approach to comprehensively assess the acute toxicity, sublethal effects, and transgenerational impacts of cyclaniliprole on *R. padi* and *S. graminum*. The results provide valuable insights for evaluating the overall insecticidal potential of cyclaniliprole and optimizing integrated pest management strategies for wheat aphids.

## 2. Materials and Methods

### 2.1. Insects and Insecticides

The populations of *R*. *padi* and *S. graminum* were collected in April 2024 from a wheat field located in Xinxiang, Henan Province, China (35°09′12″ N, 113°39′29″ E). The aphids were maintained on wheat seedlings (Aikang 58) under controlled laboratory conditions: temperature at 19–21 °C, relative humidity at 60 ± 10%, and a 16:8 h (light–dark) photoperiod. Throughout the rearing period, no pesticide exposure occurred. Cyclaniliprole (95% active ingredient, *w*/*w*) was provided by Ishihara Sangyo Kaisha (ISK) (Osaka, Japan).

### 2.2. Insecticide Bioassays

The toxicity of cyclaniliprole against *R. padi* and *S*. *graminum* was evaluated using a leaf-dip bioassay [20]. A stock solution of cyclaniliprole (10,000 mg/L) was prepared in acetone and serially diluted with 0.1% (*v*/*v*) Tween-80 aqueous solution to obtain five test concentrations: 1, 5, 25, 125, and 625 mg/L. Wheat leaves with approximately 30 third-instar wingless aphids were immersed in each test solution for 3–5 s, transferred to a Petri dishes lined with moist filter paper, and then maintained under the aforementioned conditions. Mortality was assessed 24 h post-treatment, with aphids considered dead if they showed no more than one leg movement upon gentle prodding with a dissection needle. A 0.1% (*v*/*v*) Tween-80 aqueous solution served as the untreated control, and each concentration was replicated three times. The mortality of control in every experiment request lower than 5%. Probit analysis (PoloPlus 2.0 software, LeOra Software, USA) was used to determine LC_15_, LC_35_, and LC_50_ values, along with the slope and Chi-square statistic. The derived sublethal concentrations (LC_15_ and LC_35_) were then applied to evaluate the sublethal effects of cyclaniliprole on *R. padi* and *S. graminum*.

### 2.3. Sublethal and Transgenerational Effects of Cyclaniliprole on R. padi and S. graminum

The sublethal and transgenerational effects of cyclaniliprole on *R. padi* and *S. graminum* were evaluated using life table analysis. A cohort of third-instar nymphs was treated by immersion in LC_15_ and LC_35_ solutions, as described in the bioassay section, with 0.1% (*v*/*v*) Tween-80 in water serving as the control. After 24 h, mortality was recorded, and sixty surviving wingless aphids were randomly selected and individually transferred to a plastic Petri dishes, each containing a moistened filter paper and three wheat seedlings (5–10 cm long). Three wheat seedlings (5–10 cm long) were wrapped with water-soaked cotton before being placed in 1.5 mL microtubes. Fresh wheat seedlings were replaced every three days. The longevity of F_0_ adults and their daily nymph production were recorded until all aphids died. Subsequently, sixty newborn nymphs (<24 h old) from each treatment group were randomly selected as the F_1_ generation. Each F_1_ nymph was reared individually on fresh wheat seedlings under the same conditions. The developmental duration of each life stage and daily nymph production of F_1_ individuals were monitored, with newborn nymphs removed after counting. Observations continued until all aphids died. The collected data were used to construct age-stage-specific two-sex life tables for analysis.

### 2.4. Life Table Analysis

A life table analysis of *R. padi and S. graminum* was conducted using the TWO-SEX-MSChart software (Version 2025.03.23) [21] following the age-stage two-sex life table theory [22,23]. The evaluated parameters included the developmental duration of different life stages, adult longevity, adult pre-reproductive period (APRP), total pre-reproductive period (TPRP), reproductive days, fecundity, intrinsic rate of increase (*r*), finite rate of increase (*λ*), net reproductive rate (*R*_0_), and mean generation time (*T*). These parameters were compared using a paired bootstrap test based on the confidence interval of differences [21,24]. Additionally, age-stage-specific survival rate (*sₓⱼ*), age-specific survival rate (*l_x_*), age-specific fecundity of the total population (*m_x_*), age-specific maternity (*lₓmₓ*), age-stage reproductive value (*vₓⱼ*), and age-specific life expectancy (*eₓⱼ*) were computed using the TWOSEX-MSChart program (Version 2025.03.23) [25]. The variance and standard errors of the population parameters were estimated via the bootstrap procedure incorporated in TWOSEX-MSChart, with 100,000 random resamplings [26,27].

## 3. Results

### 3.1. The Toxicity and Sublethal Concentration of Cyclaniliprole on R. padi and S. graminum

The toxicity parameters of cyclaniliprole against *R. padi* and *S. graminum* are presented in Table 1. The results show that the estimated LC_15_, LC_35_, and LC_50_ values of cyclaniliprole against *R. padi* were 4.19, 16.90, and 38.56 mg/L, respectively. Similarly, the LC_15_, LC_35_, and LC_50_ values against *S. graminum* were 2.17, 12.16, and 33.71 mg/L, respectively. For *R. padi*, exposure to 5.00 and 17.00 mg/L of cyclaniliprole (corresponding to LC_15_ and LC_35_) resulted in 13.83% and 37.62% mortality, respectively. In the case of *S. graminum*, exposure to 2.50 and 12.00 mg/L of cyclaniliprole (equivalent to LC_15_ and LC_35_) led to 17.26% and 36.92% mortality, respectively.

### 3.2. Sublethal Effects of Cyclaniliprole on the F_0_ Generation of R. padi and S. graminum

The adult longevity of the F_0_ generation of *R. padi* and *S. graminum* in the control group was 13.26 ± 0.97 and 8.40 ± 0.69 d, respectively (Figure 1). Compared to the control group, the adult longevity of the F_0_ generation of *R. padi* and *S. graminum* was reduced to 7.17 ± 1.06 d (*p* < 0.001) and 7.5 ± 0.55 d, 3.26 ± 0.83 d (*p* < 0.001), and 5.76 ± 0.46 d (*p* = 0.001), respectively, when treated with LC_15_ and LC_35_ concentrations of cyclaniliprole (Figure 1). The nymphs per female of the F_0_ generation of *R. padi* and *S. graminum* in the control group were 45.28 ± 3.93 and 20.30 ± 2.86, respectively (Figure 1). Compared to the control group, the number of nymphs per female of *R. padi* and *S. graminum* F_0_ generation decreased to 25.61 ± 4.47 (*p* = 0.001) and 17.20 ± 2.38, 11.12 ± 3.43 (*p* < 0.001), and 11.58 ± 2.00 (*p* = 0.013), respectively, after treatment with LC_15_ and LC_35_ concentrations of cyclaniliprole (Figure 1).

### 3.3. Transgenerational Sublethal Effects of Cyclaniliprole on Developmental Time and Fecundity of R. padi and S. graminum

The developmental duration and fecundity of the F_1_ generation of *R. padi* and *S. graminum*, derived from parents exposed to two sublethal concentrations of cyclaniliprole, are summarized in Table 2. For the *R. padi* F_1_ generation, no significant differences were observed in the duration of the first nymph stage, second nymph stage, third nymph stage, pre-adult period, adult longevity, total longevity, APRP, and TPRP between the control and treatment groups (*p* > 0.05); however, the LC_35_ concentration of cyclaniliprole significantly reduced the fourth nymph stage (*p* < 0.05). Additionally, the reproductive period and nymphs per female of the *R. padi* F_1_ generation in the LC_35_ treatment group were significantly lower compared to the LC_15_-treatment group (*p* < 0.05). For *S. graminum* F_1_ generation, the LC_15_ concentration of cyclaniliprole significantly reduced the duration of the third nymph stage compared to the control group (*p* < 0.05), whereas no significant differences were observed in other parameters of developmental duration and fecundity (*p* < 0.05).

### 3.4. Transgenerational Sublethal Effects of Cyclaniliprole on s_xj_, l_x_m_x_, v_xj_, and e_xj_ of R. padi and S. graminum

The *s_xj_* of *R. padi* and *S. graminum* F_1_ generation are presented in Figure 2. In the control group, the maximum *s_xj_* values for 2nd, 3rd, and 4th instar nymphs and adults were 0.73, 0.63, 0.78, and 0.93, respectively. In contrast, under the LC_15_ and LC_35_ treatments, the maximum *s_xj_* values for the corresponding stages were reduced to 0.64 and 0.64 (2nd), 0.51 and 0.49 (3rd), and 0.52 and 0.54 (4th), as well as 0.83 and 0.85 (adults), respectively, indicating that sublethal exposure (LC_15_ and LC_35_) decreased *s_xj_* of *R. padi* F_1_ generation (Figure 2). For *S. graminum* F_1_ generation, LC_15_ and LC_35_ concentrations of cyclaniliprole had no significant effects on *s_xj_* (Figure 2). Additionally, sublethal exposure (LC_15_ and LC_35_ in *R*. *padi*; LC_35_ in *S*. *graminum*) significantly reduced the *l_x_* and *l_x_m_x_*, whereas *m_x_* showed no significant difference in the F_1_ generation of either species (Figure 3).

The *v_xj_* did not differ significantly between treatment and control groups during the early developmental stages. However, in the later adult stages, the LC_15_-treated group of *R*. *padi* exhibited significantly reduced *v_xj_* compared to the control group, while the LC_35_-treated group of *S*. *graminum* showed significantly increased *v_xj_* relative to the control group (Figure 4). For *R*. *padi* F_1_ generation, the maximum *e_xj_* values for the 1st, 2nd, 3rd, and 4th instar nymphs and adults in the control group were 18.63, 19.08, 18.08, 17.08, and 16.08 d, respectively. Under LC_15_ and LC_35_ treatments, these values decreased to 16.18 and 16.77 (1st instar), 16.12 and 17.43 (2nd instar), 16.23 and 17.14 (3rd instar), and 15.25 and 16.28 (4th instar), as well as 14.45 and 15.28 days (adults), respectively, indicating that sublethal exposure reduced *e_xj_* in *R*. *padi* (Figure 5). Similarly, for *S*. *graminum* F_1_ generation, the maximum *e_xj_* values declined from 13.25, 13.38, 12.19, 11.42, and 10.66 days (control group) to 12.14, 11.09, 10.00, 10.51, and 9.51 days (LC_35_-treated group) for the 1st through to the 4th instars and adults, respectively, indicating that LC_35_ exposure significantly reduced *e_xj_* in *S*. *graminum* (Figure 5).

### 3.5. The Effects of the Cyclaniliprole on Population Parameters of the R. padi and S. graminum F_1_ Generation

The population parameters of *R. padi* and *S. graminum* F_1_ generation are presented in Table 3. For *R. padi* F_1_ generation, no significant differences were observed among the CK, LC_15_, and LC_35_ groups in the *r*, *λ*, and *T* (*p* > 0.05). However, the *R*_0_ in the LC_35_-treated group was significantly lower than that of the control group (*p* < 0.05). For *S. graminum* F_1_ generation, no significant differences were detected among the CK, LC_15_, and LC_35_ groups in *r*, *λ*, *R*_0_, and *T* (*p* > 0.05).

## 4. Discussion

Chemical control remains the primary approach for managing wheat pests. Cyclaniliprole, as a novel third-generation anthranilic diamide insecticide, has demonstrated efficacy against various pests including aphids, whiteflies, thrips, and diamondback moth [2,4]. A recent study revealed that cyclaniliprole presents an exceptionally low resistance risk and shows minimal cross-resistance in *Myzus persicae* [28]. While these findings are promising, the toxicological effects of cyclaniliprole on wheat aphid species have not been systematically investigated. In our current study, cyclaniliprole showed pronounced acute toxicity against *R. padi* and *S. graminum*, exhibiting 24 h LC_50_ values of 38.56 mg/L and 33.71 mg/L, respectively. These quantitative results substantiate that cyclaniliprole serves as an effective chemical option for wheat aphid management.

Beyond their lethal effects on target pests, chemical insecticides frequently induce significant sublethal effects due to inherent field application variability and subsequent environmental degradation processes [12]. Sublethal insecticide exposures typically diminish adult longevity and reproductive output in the F_0_ generation of pest populations. For instance, acetamiprid sublethal concentrations were shown to significantly reduce adult lifespan and fecundity in *S. graminum* and *Sitobion miscanthi* populations [7]. Our current findings reveal that sublethal exposure (LC_15_ and LC_35_ in *R*. *padi*; LC_35_ in *S*. *graminum*) significantly reduced adult longevity and fecundity in the F_0_ generation, confirming the adverse sublethal effects on these species’ parental generations. These observations are consistent with the sub-lethal effects of most pesticides on the F_0_ generation of aphids. For instance, sublethal doses of flonicamid were shown to significantly decrease adult longevity and fecundity of *R. padi* F_0_ generation [29], and similarly reduced the reproductive performance of F_0_-generation *A. gossypii* [30].

Numerous studies have demonstrated that sublethal pesticide concentrations can induce transgenerational effects on insect’s development and reproductive capacity. For example, sublethal doses of pirimicarb were found to prolong the *T* and population doubling time while reducing the *r* in *R. padi* F_1_ generation [31]. Similarly, sublethal concentrations of sulfoxaflor and afidopyropen significantly extended the preadult period and TPRP, while decreasing *r*, *R*_0_, and *λ* in *A. gossypii* F_1_ generation [16,32]. In the current study, the LC_35_ concentration of cyclaniliprole significantly reduced duration of the fourth nymphal stage and *R*_0_ in *R. padi* F_1_ generation. And the LC_15_ concentration similarly shortened the third nymphal stage duration in *S. graminum* F_1_ generation. Furthermore, sublethal concentrations (LC_15_ and LC_35_) of cyclaniliprole significantly reduced *s_xj_*, *l_x_*, *l_x_m_x_*, and *e_xj_* in the F_1_ generation of *R*. *padi*, while only LC_35_ decreased *l_x_*, *l_x_m_x_* and *e_xj_* in *S*. *graminum*. However, other life table parameters remained unaffected in both aphid species. These findings indicate that sublethal concentrations of cyclaniliprole produce pronounced adverse transgenerational effects on *R. padi* populations, while exerting comparatively milder effects on *S. graminum*.

Certain pesticides exhibit a dose-dependent dual effect on pest toxicity, where high doses typically suppress pest population development and low dose exposures may stimulate population growth in some pest species [33]. This phenomenon, termed hormesis, has been extensively documented in previous research. Specifically, sublethal concentrations of flupyradifurone were shown to significantly increase nymph production per female and the *r* in *M. persicae* F_1_ generations [34], with similar enhancements of these population parameters observed in *Metopolophium dirhodum* following sublethal imidacloprid exposure [17]. Such insecticide-induced hormesis, which stimulates pest population growth at sublethal doses, is recognized as one of the drivers of pest resurgence [35]. Notably, our findings reveal that sublethal doses of cyclaniliprole did not significantly increase nymph production per female, *r*, and *R*_0_ in either *R. padi* or *S. graminum* F_1_ generations. These results demonstrate that the sublethal dose of cyclaniliprole has no hormesis on *R. padi and S. graminum* populations.

## 5. Conclusions

In conclusion, our results indicate that cyclaniliprole have an acute toxicity against *R. padi* and *S. graminum*, with a LC_50_ of 38.56 and 33.71 mg/L at 24 h, respectively. Sublethal exposure (LC_15_ and LC_35_ in *R*. *padi*; LC_35_ in *S*. *graminum*) significantly reduced adult longevity and fecundity in the F_0_ generation. In the F_1_ generation, cyclaniliprole at LC_35_ significantly reduced the fourth nymph stage in *R*. *padi*, whereas at LC_15_, it shortened the third nymph stage duration in *S*. *graminum* compared to the control. Sublethal concentrations (LC_15_ and LC_35_) of cyclaniliprole significantly reduced *s_xj_*, *l_x_*, *l_x_m_x_*, and *e_xj_* in the F_1_ generation of *R*. *padi*, while only LC_35_ decreased *l_x_*, *l_x_m_x_* and *e_xj_* in *S*. *graminum*. Additionally, LC_35_ significantly reduced the *R*_0_ in the F_1_ generation of *R*. *padi* compared to the control. These findings suggest that cyclaniliprole exhibit notable acute toxicity against both aphid species and that sublethal concentrations adversely affected the F_0_ generation, with no observed hormetic effects in the F_1_ generations of *R. padi* and *S. graminum*. In subsequent research, it is necessary to conduct further investigations into the control effects of cyclaniliprole on different field populations of *R. padi and S. graminum* and to further evaluate the resistance risks of these pest populations to cyclaniliprole. The findings will significantly enhance our knowledge of cyclaniliprole’s application value and contribute to the optimization of integrated pest management strategies for wheat aphid control.

## Figures and Tables

**Figure 1 insects-16-00882-f001:**
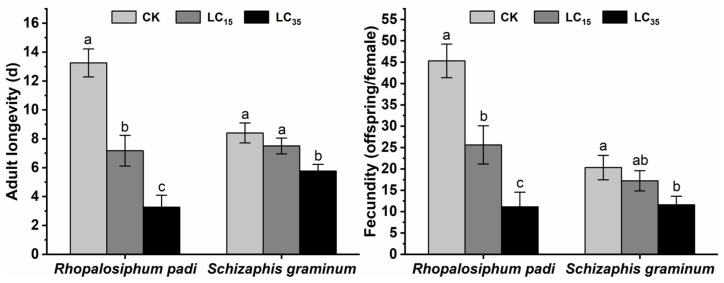
The adult longevity and fecundity of *Rhopalosiphum padi* and *Schizaphis graminum* F_0_ generation under control conditions (CKs), treated with LC_15_ and LC_35_ concentrations of cyclaniliprole. Different letters indicate significant differences (*p* < 0.05).

**Figure 2 insects-16-00882-f002:**
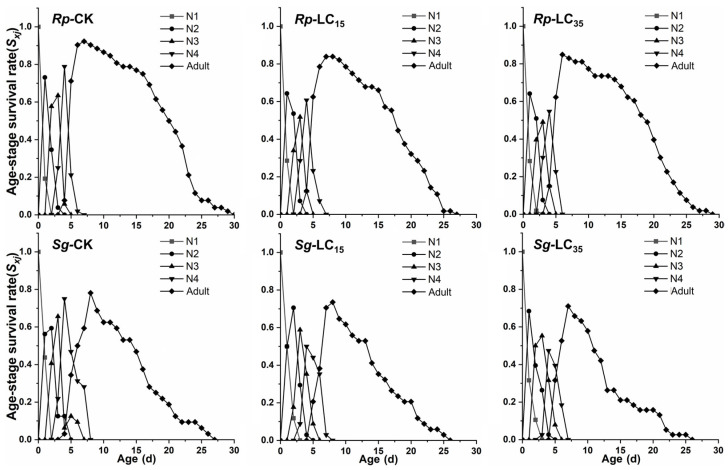
Age-stage-specific survival rate (*s_xj_*) of *Rhopalosiphum padi* and *Schizaphis graminum* F_1_ generation under control conditions (CK) treated with LC_15_ and LC_35_ concentrations of cyclaniliprole. *Rp*, *Rhopalosiphum padi*; *Sg*, *Schizaphis graminum*; N1, first nymph stage; N2, second nymph stage; N3, third nymph stage; N4, fourth nymph stage.

**Figure 3 insects-16-00882-f003:**
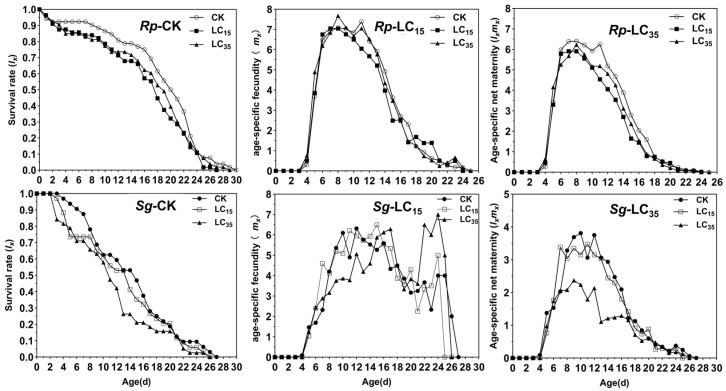
Age-specific survival rate (*l_x_*), age-specific fecundity of the total population (*m_x_*), and age-specific maternity (*l_x_m_x_*) of *Rhopalosiphum padi* and *Schizaphis graminum* F_1_ generation under control conditions (CK) treated with LC_15_ and LC_35_ concentrations of cyclaniliprole. *Rp*, *Rhopalosiphum padi*; *Sg*, *Schizaphis graminum*.

**Figure 4 insects-16-00882-f004:**
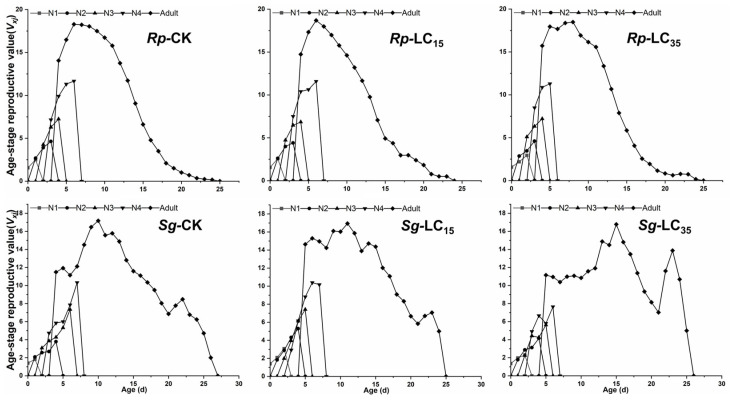
Age-stage-specific reproductive rate (*v_xj_*) of *Rhopalosiphum padi* and *Schizaphis graminum* F_1_ generation under control conditions (CK) treated with LC_15_ and LC_35_ concentrations of cyclaniliprole. *Rp*, *Rhopalosiphum padi*; *Sg*, *Schizaphis graminum*; N1, first nymph stage; N2, second nymph stage; N3, third nymph stage; N4, fourth nymph stage.

**Figure 5 insects-16-00882-f005:**
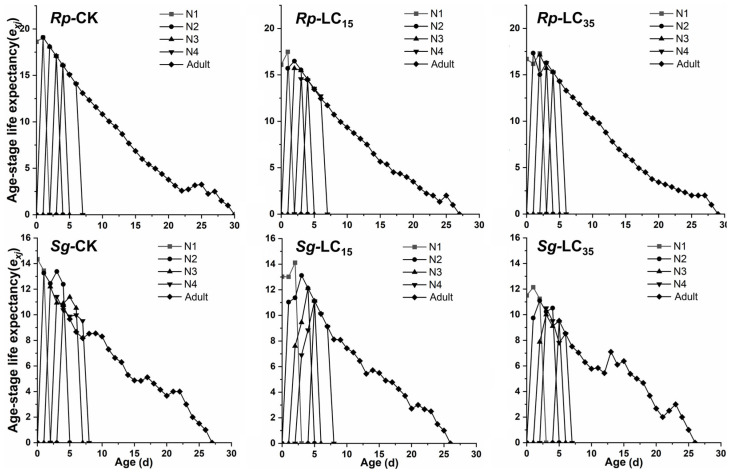
Age-stage-specific life expectancy (*e_xj_*) of *Rhopalosiphum padi* and *Schizaphis graminum* F_1_ generation under control conditions (CK), treated with LC_15_ and LC_35_ concentrations of cyclaniliprole. *Rp*, *Rhopalosiphum padi*; *Sg*, *Schizaphis graminum*; N1, first nymph stage; N2, second nymph stage; N3, third nymph stage; N4, fourth nymph stage.

**Table 1 insects-16-00882-t001:** The toxicity and sublethal concentration of cyclaniliprole on *Rhopalosiphum padi* and *Schizaphis graminum* (mg/L).

Aphids	Regression Equation	SE	Chi-Square	*p*-Value	Index	Concentration (95% CL, mg/L)	Sublethal Concentration (mg/L)	Mortality (%)
*Rhopalosiphum padi*	y = 3.29 + 1.08x	0.08	0.86	0.83	LC_15_	4.19 (2.70–5.98)	5.00	13.83
					LC_35_	16.90 (12.51–22.42)	17.00	37.62
					LC_50_	38.56 (29.05–52.23)		
*Schizaphis graminum*	y = 3.67 + 0.87x	0.10	0.77	0.86	LC_15_	2.17 (1.067–3.673)	2.5	17.26
					LC_35_	12.16 (7.56–20.00)	12.0	36.92
					LC_50_	33.71 (20.45–64.24)		

95% CL, 95% confidence limits; LC_15_, 15% lethal concentration (mg/L); LC_35_, 35% lethal concentration (mg/L); LC_50_, 50% lethal concentration (mg/L).

**Table 2 insects-16-00882-t002:** Sublethal effects of cyclaniliprole on the developmental duration and fecundity of *Rhopalosiphum padi* and *Schizaphis graminum* F_1_ generation.

Parameters	***Rhopalosiphum padi*** (Mean ± SE)			***Schizaphis graminum*** (Mean ± SE)		
	***Rp-***CK	***Rp-***LC_15_	***Rp-***LC_35_	***Sg***-CK	***Sg***-LC_15_	***Sg***-LC_35_
N1 (d)	1.44 ± 0.75 a	1.62 ± 0.12 a	1.42 ± 0.11 a	1.21 ± 0.06 a	1.31 ± 0.06 a	1.33 ± 0.07 a
N2 (d)	1.41 ± 0.12 a	1.53 ± 0.10 a	1.37 ± 0.11 a	1.21 ± 0.06 a	1.37 ± 0.07 a	1.35 ± 0.08 a
N3 (d)	1.35 ± 0.62 a	1.22 ± 0.81 a	1.25 ± 0.11 a	1.38 ± 0.07 a	1.12 ± 0.05 b	1.20 ± 0.07 ab
N4 (d)	2.17 ± 0.26 a	1.68 ± 0.13 ab	1.37 ± 0.35 b	1.37 ± 0.07 a	1.37 ± 0.07 a	1.27 ± 0.07 a
Pre-adult (d)	6.35 ± 0.25 a	6.24 ± 0.19 a	5.81 ± 0.16 a	5.17 ± 0.09 a	5.21 ± 0.11 a	5.09 ± 0.10 a
Adult longevity (d)	8.96 ± 0.90 a	9.88 ± 0.86 a	8.70 ± 0.86 a	14.92 ± 0.71 a	13.27 ± 0.71 a	14.20 ±0.77 a
Total longevity (d)	14.34 ± 1.09 a	13.03 ± 1.15 a	11.50 ± 1.0 a	18.62 ± 0.97 a	16.09 ± 0.99 a	16.70 ± 1.06 a
APRP (d)	0.03 ± 0.03 a	0.03 ± 0.03 a	0.02 ± 0.03 a	0.12 ± 0.05 a	0.15 ± 0.06 a	0.11 ± 0.05 a
TPRP (d)	6.38 ±0.24 a	6.24 ± 0.18 a	5.82 ± 0.16 a	5.30 ± 0.09 a	5.35 ± 0.12 a	5.20 ± 0.10 a
Reproductive period (d)	8.38 ± 0.92 ab	9.44 ± 0.81 a	7.22 ± 0.77 b	11.71 ± 0.55 a	10.52 ± 0.58 a	10.93 ± 0.59 a
Fecundity (nymphs per female)	42.62 ± 5.12 ab	49.78 ± 4.71 a	34.49 ± 4.82 b	68.21 ± 3.32 a	61.50 ± 3.40 a	66.62 ± 3.43 a

*Rp*, *Rhopalosiphum padi*; *Sg*, *Schizaphis graminum*; CK, control group; N1, first nymph stage; N2, second nymph stage; N3, third nymph stage; N4, fourth nymph stage; Pre-adult, complete nymph stage; APRP, adult pre-reproductive period; TPRP, total pre-reproductive period. Data are presented as the mean ± SE. Values in the same row followed by different letters are significantly different (*p* < 0.05) from the respective control values.

**Table 3 insects-16-00882-t003:** Sublethal effects of cyclaniliprole on population parameters of the F_1_ generation of *Rhopalosiphum padi* and *Schizaphis graminum*.

Index	***Rhopalosiphum padi*** (Mean ± SE)			***Schizaphis graminum*** (Mean ± SE)		
	***Rp-***CK	***Rp-***LC_15_	***Rp-***LC_35_	***Sg***-CK	***Sg***-LC_15_	***Sg***-LC_35_
*r*	0.3404 ± 0.0109 a	0.3371 ± 0.0133 a	0.3056 ± 0.01501 a	0.4462 ± 0.0092 a	0.4393 ± 0.0115 a	0.4484 ± 0.0125 a
*λ*	1.4055 ± 0.0153 a	1.4008 ± 0.0186 a	1.3575 ± 0.0203 a	1.5620 ± 0.0144 a	1.5518 ± 0.0179 a	1.5652 ± 0.0196 a
*R* _0_	38.62 ± 5.14 a	36.59 ±5.12 ab	24.52 ± 4.26 b	62.97 ± 3.98 a	52.72 ± 4.08 a	56.56 ± 4.38 a
*T*	10.73 ± 0.38 a	10.68 ± 0.38 a	10.47 ± 0.47 a	9.29 ± 0.14 a	9.02 ± 0.16 a	9.01 ± 0.18 a

*Rp*, *Rhopalosiphum padi*; *Sg*, *Schizaphis graminum*; CK, control group; *r*, intrinsic rate of increase (%); *λ*, finite rate of increase (%); *R*_0_, net reproductive rate (%); *T*, mean generation time (d). Data are presented as the mean ± SE. Values in the same row followed by different letters are significantly different (*p* < 0.05) from the respective control values.

## Data Availability

The original contributions presented in this study are included in the article. Further inquiries can be directed to the corresponding authors.

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
