# Peer review of "Sublethal and Transgenerational Effects of Cyclaniliprole on Demographic Parameters in *Rhopalosiphum padi* and *Schizaphis graminum* (Hemiptera: Aphididae)"

_insects, 2025, doi:10.3390/insects16090882_

Round 1
Reviewer 1 Report
Comments and Suggestions for Authors
It is recommended to reduce the similarity percentage of your work to a minimum of 15%.
The methodology is appropriate; a breeding preparation is presented; trials were conducted to test the product's insecticidal capacity and evaluate sublethal and transgenerational effects. Future follow-up work could be recommended to implement greenhouse and/or field studies to scale up the results and expand the data to estimate whether resistance may develop in these populations in the short term and what impacts it would have.
Author Response
Reviewer: 1
Comments and Suggestions for Authors
The methodology is appropriate; a breeding preparation is presented; trials were conducted to test the product's insecticidal capacity and evaluate sublethal and transgenerational effects. Future follow-up work could be recommended to implement greenhouse and/or field studies to scale up the results and expand the data to estimate whether resistance may develop in these populations in the short term and what impacts it would have.
Author’s Response: Thanks. We will conduct in-depth research on these work in the next step.
Reviewer 2 Report
Comments and Suggestions for Authors
The reviewed article addresses an important and relevant topic: the toxicity of the insecticide cyclaniliprole against two major wheat pests, Rhopalosiphum padi and Schizaphis graminum. The study presents results on both the lethal effects of the compound and its sublethal and transgenerational impacts. These findings have the potential to contribute valuable information for the development of effective crop protection strategies.
However, the manuscript contains several significant methodological flaws and interpretative inconsistencies that undermine the reliability of the results and compromise the conclusions drawn.
The authors employed the age-stage two-sex life table method to analyze the data. While this approach is commonly used, it is not well-suited to insects that reproduce primarily through parthenogenesis. Aphids, the subject of this study, give birth to live nymphs under controlled laboratory conditions. Therefore, references to sex-specific parameters and oviposition-related traits are methodologically unjustified and lead to misinterpretation of the results.
Moreover, the Materials and Methods section lacks essential information regarding the experimental conditions. The authors do not specify the temperature, humidity, or light regime used during the experiments. Several sections contain ambiguous or contradictory details. For instance, in Section 2.2, it is unclear whether aphids were transferred to Petri dishes with or without host plants after being immersed in the treatment solution. In Section 2.3, it is stated that aphids were reared individually, yet elsewhere the text mentions the use of three wheat seedlings. Furthermore, there is no information about the month in which aphids were collected from the field, which is an important detail that reflects the authors’ understanding of aphid biology and seasonal dynamics.
In summary, the use of demographic parameters related to sex and oviposition in a parthenogenetic species under controlled laboratory conditions, combined with the lack of clear methodological description, significantly reduces the scientific credibility of the manuscript. In my opinion, the study should be revised and improved substantially. The results should be based on reliable empirical data, not on simulations generated by statistical software without adequate biological justification.
Therefore, I recommend rejection of the manuscript in its current form.
Author Response
Author's Notes
Reviewer: 2
Comments and Suggestions for Authors
The reviewed article addresses an important and relevant topic: the toxicity of the insecticide cyclaniliprole against two major wheat pests, Rhopalosiphum padi and Schizaphis graminum. The study presents results on both the lethal effects of the compound and its sublethal and transgenerational impacts. These findings have the potential to contribute valuable information for the development of effective crop protection strategies.
Author’s Response: Thanks.
However, the manuscript contains several significant methodological flaws and interpretative inconsistencies that undermine the reliability of the results and compromise the conclusions drawn.
Author’s Response: We made careful revisions to methodological flaws and interpretative inconsistencies in manuscript.
The authors employed the age-stage two-sex life table method to analyze the data. While this approach is commonly used, it is not well-suited to insects that reproduce primarily through parthenogenesis. Aphids, the subject of this study, give birth to live nymphs under controlled laboratory conditions. Therefore, references to sex-specific parameters and oviposition-related traits are methodologically unjustified and lead to misinterpretation of the results.
Author’s Response: We referred to a large number of studies on the sub-lethal effects of pesticides on aphids, so we adopted the age-stage two-sex life table method to analyze the data (Shang et al. 2021, Ma et al. 2022, Shi et al. 2022, Gul et al. 2024). And we have changed "female" to "adult", changed " oviposition period" to " reproductive period" in the manuscript.
Gul, H., A. Güncan, F. Ullah, N. Desneux, and X. Liu. 2024. Intergenerational Sublethal Effects of Flonicamid on Cotton Aphid, Aphis gossypii: An Age-Stage, Two-Sex Life Table Study. Insects 15.
Ma, K., Q. Tang, P. Liang, J. Li, and X. Gao. 2022. A sublethal concentration of afidopyropen suppresses the population growth of the cotton aphid, Aphis gossypii Glover (Hemiptera: Aphididae). J. Integr. Agr. 21: 2055-2064.
Shang, J., Y. S. Yao, X. Z. Zhu, L. Wang, D. Y. Li, K. X. Zhang, X. K. Gao, C. C. Wu, L. Niu, J. C. Ji, J. Y. Luo, and J. J. Cui. 2021. Evaluation of sublethal and transgenerational effects of sulfoxaflor on Aphis gossypii via life table parameters and 16S rRNA sequencing. Pest Manag. Sci. 77: 3406-3418.
Shi, D., C. Luo, H. Lv, L. Zhang, N. Desneux, H. You, J. Li, F. Ullah, and K. Ma. 2022. Impact of sublethal and low lethal concentrations of flonicamid on key biological traits and population growth associated genes in melon aphid, Aphis gossypii Glover. Crop Prot. 152.
Moreover, the Materials and Methods section lacks essential information regarding the experimental conditions. The authors do not specify the temperature, humidity, or light regime used during the experiments. Several sections contain ambiguous or contradictory details. For instance, in Section 2.2, it is unclear whether aphids were transferred to Petri dishes with or without host plants after being immersed in the treatment solution. In Section 2.3, it is stated that aphids were reared individually, yet elsewhere the text mentions the use of three wheat seedlings. Furthermore, there is no information about the month in which aphids were collected from the field, which is an important detail that reflects the authors’ understanding of aphid biology and seasonal dynamics.
Author’s Response: We made careful revisions to lacks essential information regarding the experimental conditions in the materials and methods section of manuscript.
In summary, the use of demographic parameters related to sex and oviposition in a parthenogenetic species under controlled laboratory conditions, combined with the lack of clear methodological description, significantly reduces the scientific credibility of the manuscript. In my opinion, the study should be revised and improved substantially. The results should be based on reliable empirical data, not on simulations generated by statistical software without adequate biological justification.
Author’s Response: Thank you very much for your excellent suggestion. We made careful revisions to manuscript.
Therefore, I recommend rejection of the manuscript in its current form.
Author’s Response: We made careful revisions to manuscript.
Reviewer 3 Report
Comments and Suggestions for Authors
The manuscript evaluates the sublethal and transgenerational effects of cyclaniliprole on Rhopalosiphum padi and Schizaphis graminum on wheat. A significant reduction of longevity and fecundity on aphids (F0) of both species treated with LC35, as well as significant transgenerational (F1) reductions in duration of some developmental periods, fecundity and net reproductive rate of R. padi but not S. graminum was found. The manuscript is well written with a complete and updated literature review. The results obtained support their main conclusions, with standard bioassays and statistical analyses. Some suggestions and minor comments are provided for the improvement of the manuscript.
- The title should describe better the main results. Instead of “biological traits” that is too broad, you should name fecundity of population growth rate. Furthermore, please remove “2 species of wheat aphids” because they are already named.
- Lines 90-95: Age-stage two-sex life table approach is not so important in this study to be mentioned in the introduction. Wheat aphids were maintained in conditions that ensure asexual reproduction and all experiments were run with viviparous females. Conventional life tables should have provided the same results.
- Line 108: Please italics in scientific names along the manuscript.
- Lines 132-133: Do you obtained any alate adults in the experiments? If so, were they included in the analyses? Fecundity between aphid morphs (apterous vs alates) is very different.
- Line 143: adult longevity is not mentioned here…
- Lines 160-161: This sentence with interpretation of results is not necessary here and is better to include it in the discussion section.
- Table 1: Please also include estimation of intercept and SE in the Probit table.
- Lines 167-177: Please do not include the values of non-significant results in this paragraph. In particular, the phrase “and 7.5 ± 0.55 d (P= 0.255),” in line 170 and “and 17.20 ± 2.38 (P= 0.371),” in line 175.
- Lines 177-179: This sentence with interpretation of results is not necessary here and is better to include it in the discussion section.
- Figure 1: Line 183: Please include control abbreviation CK in the legend.
- Figures 2-5: Lines 201-218. It is difficult just looking at the figures to see if there are significant differences or not in age specific variables. Are differences found at all ages in all variables? More details about statistical comparisons between age specific variables are necessary. Perhaps supplementary tables with more details could be included.
- Table 2: Line220: Please provide units of stage duration and fecundity for all variables in the table or legend. Include abbreviation of control treatment.
- Figure 2-5: Legends of Y axis is redundant in all graphs of the same figure. Legend of X axis is redundant in Figure 4 and 5.
- Line 320: Conclusion: Please do not use the verb “demonstrate”, which is more appropriate for exact sciences. I suggest to use the verbal forms “support” or “provides evidence in support of”… even the simple description that cyantraniliprole exhibits toxicity is better.
- Lines 321-323: Conclusion: Please provide further details like in the abstract. Strictly speaking this is not right for LC15 on S. graminum. Please also describe the transgenerational effects found for each species.
- There are scientific names without italics or capital letter in the genus name in several references. Titles of articles are with and without capital letters in different references. A few journal names are not abbreviated.
Author Response
Reviewer: 3
Comments and Suggestions for Authors
The manuscript evaluates the sublethal and transgenerational effects of cyclaniliprole on Rhopalosiphum padi and Schizaphis graminum on wheat. A significant reduction of longevity and fecundity on aphids (F0) of both species treated with LC35, as well as significant transgenerational (F1) reductions in duration of some developmental periods, fecundity and net reproductive rate of R. padi but not S. graminum was found. The manuscript is well written with a complete and updated literature review. The results obtained support their main conclusions, with standard bioassays and statistical analyses. Some suggestions and minor comments are provided for the improvement of the manuscript.
Author’s Response: Thanks.
The title should describe better the main results. Instead of “biological traits” that is too broad, you should name fecundity of population growth rate. Furthermore, please remove “2 species of wheat aphids” because they are already named.
Author’s Response: We have revised in title.
Lines 90-95: Age-stage two-sex life table approach is not so important in this study to be mentioned in the introduction. Wheat aphids were maintained in conditions that ensure asexual reproduction and all experiments were run with viviparous females. Conventional life tables should have provided the same results.
Author’s Response: We have removed these contents.
Line 108: Please italics in scientific names along the manuscript.
Author’s Response: We have revised in line 106.
Lines 132-133: Do you obtained any alate adults in the experiments? If so, were they included in the analyses? Fecundity between aphid morphs (apterous vs alates) is very different.
Author’s Response: We did not obtain any alate adults in the experiments. We selected wingless aphids for the experiment (line 125).
Line 143: adult longevity is not mentioned here…
Author’s Response: We have revised in line 140.
Lines 160-161: This sentence with interpretation of results is not necessary here and is better to include it in the discussion section.
Author’s Response: We removed this sentence and include it in the discussion section. Line 159.
Table 1: Please also include estimation of intercept and SE in the Probit table.
Author’s Response: We have revised in table 1.
Lines 167-177: Please do not include the values of non-significant results in this paragraph. In particular, the phrase “and 7.5 ± 0.55 d (P= 0.255),” in line 170 and “and 17.20 ± 2.38 (P= 0.371),” in line 175.
Author’s Response: We have revised in line 169 and line 174.
Lines 177-179: This sentence with interpretation of results is not necessary here and is better to include it in the discussion section.
Author’s Response: We removed this sentence and include it in the discussion section. Line 175
Figure 1: Line 183: Please include control abbreviation CK in the legend.
Author’s Response: We have revised in the legend of figure 1. Line 178.
Figures 2-5: Lines 201-218. It is difficult just looking at the figures to see if there are significant differences or not in age specific variables. Are differences found at all ages in all variables? More details about statistical comparisons between age specific variables are necessary. Perhaps supplementary tables with more details could be included.
Author’s Response: We have revised in line 196-218.
Table 2: Line220: Please provide units of stage duration and fecundity for all variables in the table or legend. Include abbreviation of control treatment.
Author’s Response: We have revised in table 2.
Figure 2-5: Legends of Y axis is redundant in all graphs of the same figure. Legend of X axis is redundant in Figure 4 and 5.
Author’s Response: We have revised in Figure 2-5.
Line 320: Conclusion: Please do not use the verb “demonstrate”, which is more appropriate for exact sciences. I suggest to use the verbal forms “support” or “provides evidence in support of”… even the simple description that cyantraniliprole exhibits toxicity is better.
Author’s Response: We have revised in line 318.
Lines 321-323: Conclusion: Please provide further details like in the abstract. Strictly speaking this is not right for LC15 on S. graminum. Please also describe the transgenerational effects found for each species.
Author’s Response: We have revised in line 318-326.
There are scientific names without italics or capital letter in the genus name in several references. Titles of articles are with and without capital letters in different references. A few journal names are not abbreviated.
Author’s Response: We carefully checked and revised the reference citation format.
Round 2
Reviewer 2 Report
Comments and Suggestions for Authors
I do not revise my previous critical assessment of the manuscript. I maintain that the applied method is inappropriate for aphids. The authors continue to report the oviposition period in their results, which remains biologically unjustified. I leave the decision regarding further processing of the manuscript to the editors.